# **Cross-calibration of GOME and SCIAMACHY Spectrometers Enhanced by Polarization Monitoring Devices Data**

Abdalmenem Owda <sup>1</sup>, Melanie Coldewey-Egbers <sup>1</sup>, Sander Slijkhuis <sup>1</sup>, Günter Lichtenberg <sup>1</sup>, and Bernd Aberle <sup>1</sup>

<sup>1</sup>Remote Sensing Technology Institute, German aerospace center (DLR), Weßling, Germany

**Correspondence:** Abdalmenem Owda (abdalmenem.owda@dlr.de)

**Abstract.** Spectrometer instruments have significantly contributed to monitoring atmospheric composition and climate change for decades. Among them, the Global Ozone Monitoring Experiment (GOME, 1995–2011) and the Scanning Imaging Absorption Spectrometer for Atmospheric Chartography (SCIAMACHY, 2002–2012) were two well-known sensors whose missions overlapped by nearly a decade. Both instruments provided valuable data for atmospheric applications. However, to ensure data consistency and extend long-term time series, cross-calibration between the two instruments was important. The Fundamental Data Record for Atmospheric Composition (FDR4ATMOS) project, initiated by the European Space Agency (ESA), aims at harmonizing GOME and SCIMACHY Level 1 data, i.e., irradiance and reflectance measurements.

This paper presents, for the first time, the cross-calibration methodology for spectrometers used in the FDR4ATMOS project. Several challenges, such as differing spatial resolutions, lack of exact spatiotemporal overlap, and the need to preserve spectral structure, were addressed using targeted strategies. This process involved selecting scenes with minimal acquisition time differences over Pseudo-Invariant Calibration Sites (PICS) characterized by stable meteorological and atmospheric conditions. A key step involves spatially weighted averaging of SCIAMACHY pixels within each GOME footprint and computing spectral channel-wise ratios over Bands 2B, 3, and 4, which represent ultraviolet, visible, and near-infrared (UV/VIS/NIR) wavelengths. Additionally, the paper presented an analysis approach based on Polarization Monitoring Devices (PMDs) data to investigate the spatial homogeneity of pixels used in the cross-calibration and its influence on the performance of the cross-calibration.

Observations under near-clear-sky conditions from 2003 were collocated over PICS and used to derive transfer functions (TFs). Polynomial TFs were fitted for Bands 2B and 3, while a constant TF was used for Band 4. The TFs showed dependence on viewing zenith angle (VZA), degradation, and wavelength. The uncertainty of TFs increased with wavelength, corresponding to reduced homogeneity in PMD measurements. Using PMD measurements from cross-calibrated pixels as an indicator to filter out non-homogeneous pixels of the main spectral channels resulted in an uncertainty reduction up to 70% in the TFs.

Overall, the presented cross-calibration approach and PMD-based analysis provide a pathway toward generating consistent and long-term spectrometer records. This work highlights the potential for expanding future TFs derivation beyond ideally suited scenes, increasing robustness across the varied surface and atmospheric conditions.

#### 1 Introduction

#### 1.1 Motivation

25

Satellite remote sensing data has been increasingly used in various operational applications, such as atmospheric applications (Zhang et al., 2020). The increased number of satellite missions has led to the rapid growth of remote sensing archives from diverse satellite missions. These archives help to track and monitor parameters and phenomena. In addition to high spatial and radiometric resolution, they offer global coverage. One of these applications, which received considerable attention in the last decades, is the retrieval of trace gases (Abad et al., 2019; He et al., 2024). The focus arises from the urgent need to address atmospheric change and its impacts, aiming to make accurate emissions monitoring essential for effective decision-making in air quality.

Major trend analyses regarding the atmospheric environment require long-term datasets to provide a continuous and comprehensive view of gases. The time series analysis should have more than a decade of data, which could be greater than the individual satellite lifetime (Cabor et al., 1998). To avoid discontinuity in time series data, it is necessary to fully use the ever-increasing number of satellite systems from the historical and current missions. To generate consistent time series data from multi-mission sensors, several factors should be considered, including the differences in the sensors' characteristics, such as over-passing time, data quantity, viewing geometry, solar geometry, instrument-related effects, and atmospheric conditions (Barnes et al., 2021).

## 40 1.2 Cross-calibration based on Pseudo-Invariant Calibration Sites

Cross-calibration is a method of ensuring the consistency, accuracy, and comparability of measurements from different instruments, platforms, and missions. Cross-calibration consists of comparing the monitored measurements to reference instruments (Chander et al., 2013b). For trace gas applications, monitored parameters include reflectance or radiance and solar irradiance. The optimal situation to perform the cross-calibration is to acquire the measurements from both sensors simultaneously in the same geometrical and solar conditions. However, the spatiotemporal coincidence measurements are usually not very frequent, which complicates the applications of the cross-calibration.

Vicarious calibration is one of the well-known methods of absolute calibration based on precise measurements of spectral reflectance from ground instruments over a target (Patel et al., 2016; Wang et al., 2025), or measurements acquired over sites with invariant surface properties such as deserts or oceans (Ma et al., 2015). Many other man-made features, such as concrete structures, preserve the ground reflectance values and do not change significantly over time. These features, known as pseudo-invariant targets, can also serve as reference measurements for calibration (Hadjimitsis et al., 2009). This type of cross-calibration is independent of the satellites' onboard calibration systems.

Pseudo-Invariant Calibration Sites (PICS) are one of the most widely used sites in the vicarious calibration methods, e.g. (Wu et al., 2020; Barsi et al., 2016; Cabor et al., 1998; Helder et al., 2013; Chander et al., 2013a). (Cosnefroy et al., 1996) proposed 20 desert sites in Africa and the Arabian Peninsula with  $100 \times 100$  km2. (Cosnefroy et al., 1996) demonstrated the methods and criteria used for identifying PICS based on observations taken from Meteosat-4 from July 1989 to January

80

1990. The selection criteria ranged from spatial and temporal variability of the surface reflectance, cloud cover, precipitation, geomorphology, and bidirectional effects. It has been found that these sites had spatial uniformity better than 3% in a multi-temporal series of cloud-free images. The primary characteristics of these sites are limited precipitation, a lack of vegetation cover, and minimal human-induced changes. Subsequently, it has been assumed in many studies that the PICS are invariant in time (Khadka et al., 2021), and the radiance upwelling from the Earth can be assumed to be constant (Helder et al., 2013). Therefore, the main part of the changes observed in the temporal stability of sites will be directly related to the changes that occurred to the sensors.

## 1.3 GOME and SCIAMACHY spectrometers cross-calibration

The Global Ozone Monitoring Experiment (GOME) and Scanning Imaging Absorption Spectrometer for Atmospheric Cartography (SCIAMACHY) were two spectrometers used primarily for the retrieval of trace gases (see Section 2). (Owda and Lichtenberg, 2025) compared the life mission reflectance data of GOME and SCIAMACHY over the PICS for near-clear-sky conditions. They found that the reflectance time series of SCIAMACHY observations were more uniform and stationary compared with the GOME reflectance time series. The same study showed that GOME reflectance had severe degradation in 2001 and toward the end of its life mission, in agreement with (Coldewey-Egbers et al., 2018). Therefore, it was recommended to use SCIAMACHY as the reference sensor in the cross-calibration. The SCIAMACHY data from 2003 are selected as a reference to perform the harmonization between GOME and SCIAMACHY. This choice is due to (i) SCIAMACHY being at the beginning of the mission and exhibiting no or negligible instrument degradation. The reference point for the SCIAMACHY degradation was taken as February 27th, 2003. Therefore, choosing the same year minimizes degradation-related uncertainties.

(ii) SCIAMACHY had better capabilities for in-flight calibration, and Level 1b data have better degradation correction (Hilbig et al., 2019; Lichtenberg et al., 2006).

The reflectance and solar irradiance from Level 1 of GOME and SCIAMACHY have recently been cross-calibrated in the Fundamental Data Records in the Domain of Satellite Atmospheric Composition (FDR4ATMOS) project. The focus of the FDR4ATMOS was on three spectral windows in the: ultra-violet, visible, and near-infrared (UV/VIS/NIR)(see Table 2).

The cross-calibration between the satellite spectrometers GOME and SCIAMACHY presents several unique challenges. Despite the significant temporal overlap of the two missions, there were no simultaneous observations at the same locations. Additionally, differences in spatial resolution and the necessity to preserve the spectral structure of the observations further complicate the cross-calibration process. These challenges distinguish spectrometer cross-calibration from that of imaging satellite instruments, which typically do not face such constraints. Imaging sensors generally have much higher spatial resolution and do not require strict preservation of spectral structure, allowing for more straightforward alignment methods. In contrast, spectrometers like GOME and SCIAMACHY, with their high spectral but lower spatial resolution, require specialized approaches that maintain the integrity of spectral data throughout the calibration process.

This paper focuses on addressing these issues, proposing a methodology tailored specifically for cross-calibrating satellite spectrometers, rather than conventional imaging instruments. The paper is structured as follows: Sections 2 and 3 describe the

sensors and study areas. The cross-calibration method is presented in Section 4. Sections 5 and 6 present the results and the discussions, respectively. Finally, the conclusion and final remarks are given in Section 7.

## 2 Sensor and data descriptions

## **2.1 GOME**

GOME (1995-2011) was a passive nadir-viewing, grating spectrometer, covering the UV, VIS, and NIR spectral ranges. It was mounted on the second European Remote Sensing (ERS-2) satellite (ESA, 2025a). GOME could provide global coverage every 3 days. The primary objective of the mission was to collect observations to measure the total column of ozone, nitrogen dioxide, clouds, aerosols, pollutants, and other atmospheric compositions. The mission measured the upwelling solar radiation reflected or scattered in the Earth's atmosphere and surface in the wavelength range of 240 nm to 790 nm.

GOME operated using both forward and backward scanning modes, producing four ground pixels per scan cycle (3 forward and 1 backward), each with an integration time of 1.5 seconds. In the forward scan, three pixels were recorded in the east, nadir, and west directions, each covering a minimum spatial resolution of  $40 \times 320 \text{ km}^2$ . The backward scan produced one pixel with a coarser spatial resolution of  $40 \times 960 \text{ km}^2$ . The instrument was equipped with three Polarization Monitoring Devices (PMDs) and a spectrometer comprising four channels. The spectral resolution was approximately 0.2 nm in the UV range and 0.4 nm in the VIS range. The PMDs are broadband detectors specifically designed to capture polarization information with high temporal resolution (Burrows et al., 1999).

#### 2.2 SCIAMACHY

SCIAMACHY (2002–2012) was a passive imaging spectrometer aboard the ENVISAT satellite (ESA, 2025b). It operated across the UV, VIS, NIR, and shortwave infrared (SWIR) spectral regions, covering wavelengths from 240 nm to 2400 nm. The instrument featured three distinct observation modes: nadir, limb, and solar/lunar occultation.

The primary mission objective was to monitor various processes that affect the chemical composition of the troposphere and stratosphere. In nadir mode, SCIAMACHY provided measurements at a spatial resolution between  $26 \times 30 \text{ km}^2$  and  $32 \times 233 \text{ km}^2$ , depending on the wavelength and orbit phase. The swath width was 960 km. The instrument was equipped with eight spectral channels and seven PMDs. Various atmospheric gases, aerosol characteristics, radiation budgets, and cloud properties were retrieved (Bovensmann et al., 1999).

The differences between GOME and SCIAMACHY are illustrated in Table 1.

**Table 1.** Spectral windows selected for FDR4ATMOS and used for harmonization. For the NIR, the  $O_2$  A-band is excluded for the TFs FDR4ATMOS manual (2024).

| Spectral window | GOME-1 | SCIAMACHY | Wavelength interval(s) |  |  |  |
|-----------------|--------|-----------|------------------------|--|--|--|
|                 | band   | band(s)   | (nm)                   |  |  |  |
| UV              | 2B     | 09 & 10   | 313–347                |  |  |  |
| VIS             | 3      | 15        | 424–495                |  |  |  |
| NIR             | 4      | 26        | 756–757 & 773–774      |  |  |  |

#### 2.3 GOME and SCIAMACHY data

The fully calibrated Level 1 products from GOME (Level 1b) and SCIAMACHY (Level 1c) are used as input for the cross-calibration method.

GOME Level 1b and SCIAMACHY Level 1c data have solar irradiance and radiance measurements, to compute reflectance, as explained in Section 4.1.1. They also have cloud parameters and sun-normalized PMD data. However, the corresponding Level 1b SCIAMACHY needs further pre-processing due to the concept of cluster data. The cluster concepts refer to a subdivision of a channel containing a specific wavelength region and detector exposure time, aiming to identify certain important spectral windows in the data. This leads to spectral bands having different integration times. The cluster integration times can also vary over one orbit.

While GOME and SCIAMACHY employ the same scanning strategy, the cluster concept results in varying pixel sizes, depending on the orbit phase and spectral range. The Level 1b SCIAMACHY data are scaled to the shortest integration time in the cluster and are calibrated with the *scial1c* tool developed by DLR (the link to the tool is under Section 8).

The following are the main parameters from both sensors used in this research study:

- 1) Reflectances: This denotes the proportion of incident solar radiation that is reflected toward the sensor. It includes contributions from both the Earth's surface and the atmosphere through scattering. The reflectance measurements are denoted as  $(R_g)$  and  $(R_s)$ , for GOME and SCIAMACHY, respectively. Reflectance served as a key parameter for the derivation of the transfer functions (TFs), which were used to update the GOME observations based on SCIAMACHY observations, as defined in Section 4.1.3.
- 2) Cloud fraction (CF): Represented by a range from 0 to 1, where 0 indicates clear sky (no clouds) and 1 indicates full cloud cover. The cloud fraction is used to assess the cloud conditions of overlapping pixels from both sensors.
  - 3) Viewing zenith angle (VZA): The VZA is the angle between the incoming radiation directed toward the sensor and the vertical line (nadir) directly below the sensor. GOME and SCIAMACHY measured the incoming radiation based on a scanning mirror that sweeps the Earth's surface in the east-west direction. The influence of the line-of-sight on the TFs is investigated using the VZA.
- 4) Information from PMDs: PMDs offer a higher spatial resolution compared to the reflectance, allowing multiple PMD measurements to be captured within a single GOME or SCIAMACHY ground pixel. This enables the investigation of variation

**Figure 1.** Selected PICS for cross-calibration include Sudan1 (28.22°E, 21.74°N), Arabia2 (50.96°E, 20.13°N), and Libya4 (23.39°E, 28.55°N), which were identified as the most stable sites among those proposed by Cosnefroy et al. (1996), according to (Owda and Lichtenberg, 2025).

at the sub-pixel scale. As a result, PMDs are utilized to assess the homogeneity of the overlapping pixels from GOME and SCIAMACHY. PMD measurements used in this study were normalized with the PMD measurements of the sun.

## 3 Study areas description

Before cross-calibration, the temporal stability of PICS and the surrounding areas was investigated to identify the most suitable sites. Large areas beyond the PICS themselves were considered in the investigation, due to the large spatial extent of GOME and SCIAMACHY pixels. A decade of SCIAMACHY observations over the 20 PICS was analysed as part of a companion study (Owda and Lichtenberg, 2025). This analysis indicated that Sudan1, Arabia2, and Libya4 were among the most stable PICS. These sites are shown in Fig. 1, and their 2003 reflectance data were used in the cross-calibration.

## 150 4 Methods

This section outlines the methods used for deriving TFs to cross-calibrate instruments (Section 4.1) and the PMD-based analysis of pixel homogeneity and cross-calibration performance (Section 4.2).

#### 4.1 Transfer functions derivation

The FDR4ATMOS cross-calibration workflow is illustrated in Fig. 2. The diagram illustrates the steps used to perform the cross-calibration on  $R_s$  and  $R_g$ , ranging from: (i) spatial collocation of SCIAMACHY and GOME scenes over the selected sites, (ii) average the spatially higher resolved SCIAMACHY over the GOME ground pixel for the overlapped scenes, yielding

Figure 2. The FDR4ATMOS workflow of the reflectance cross-calibration of GOME and SCIAMACHY. The transfer functions are calculated for UV/VIS and NIR, except the  $O_2$  A-Band (FDR4ATMOS manual, 2024).

 $R_{scia2gome}$  (iv) re-gridding the data, and compute the ratio of  $R_{scia2gome}/R_g$  for the investigated spectral ranges and the selected sites.

#### 4.1.1 GOME and SCIAMACHY reflectance

GOME measured radiance but had a randomly changing residual etalon, which cancels out in the reflectances. Therefore, using reflectances as the basis for the harmonization is beneficial. The reflectance for a particular observation is defined as:

$$R(\lambda) = \frac{\pi . I_{earth}(\lambda)}{\cos \phi_o . E_{sun}(\lambda))} \tag{1}$$

where  $R(\lambda)$  is the reflectance,  $I_{earth}(\lambda)$  is the calibrated radiance as measured by the instrument,  $\phi_o$  is the solar zenith angle, and  $E_{sun}(\lambda)$  is the solar irradiance as measured by the instrument.

# 165 4.1.2 Spatial GOME and SCIAMACHY scenes collocation

The complete GOME and SCIAMACHY archives of 2003, with CF less than 0.25, were spatially collocated. The collocation was taken as the complete scanline within the PICS center (see Fig. 4). GOME pixels were larger than SCIAMACHY pixels and the dimension of PICS. A GOME pixel could contain 1-2 SCIAMACHY scanlines. Larger areas than those of the PICS had to be considered due to the large footprint of GOME pixels. The time difference between GOME and SCIAMACHY was about 30 minutes. Fig. 3 shows the GOME and SCIAMACHY footprints.

The overlapped fractions of the intersected GOME and SCIAMACHY were calculated. Based on the overlapped fractions, the weighted mean reflectances of all SCIAMACHY ground pixels inside the GOME pixel were computed based on Equation (2)

**Figure 3.** Footprints of GOME (blue) and SCIAMACHY pixels (green). For example, for two GOME pixels, 24 intersecting SCIAMACHY pixels were identified.

$$R_{scia2gome} = \frac{\sum_{gpscia} R_{scia,i}.w_i}{\sum_{gpscia} w_i}$$
 (2)

where  $R_{scia2gome}$  is pseudo signal SCIAMACHY reflectance averaged over GOME ground pixel,  $\sum_{gpscia}$  is the summation of all SCIAMACHY ground pixels that have overlapped with the GOME ground pixel,  $R_{scia,i}$  is the original SCIAMACHY reflectance of pixel i, and  $w_i$  is the fraction of SCIAMACHY ground pixel i that lies within the GOME ground pixel.

## 4.1.3 Reflectance harmonization

The reflectances were used as the basis for the harmonization process. The differences in  $R_s$  and  $R_g$  might be due to : (i) the differences inherent in the observation geometry of the instrument, (ii) imperfect calibration, (iii) degradation of sensors, (iv) scene dependencies.

The generic formula of the harmonization can be written as follows (FDR4ATMOS manual (2024)):

$$R_{inst1} = R_{inst2} \cdot C_{\delta} \cdot C_{1,scene} (geometry, R_{inst1,2,...}) + C_{2,scene}$$

$$\tag{3}$$

where  $R_{inst1}$  and  $R_{inst2}$  are the reflectance signals of instruments 1 and 2, respectively.  $C_{\delta}$  is the correction factor for multiplicative differences between instruments.  $C_{1,scene}(geometry, R_{inst1,2,...})$  is the correction factor for scene dependent effects, and  $C_{2,scene}$  is the correction scene dependent offsets.

The  $R_g$  in Equation (4) is the GOME reflectance re-gridded to the SCIAMACHY wavelength grid  $(\lambda_s)$  from the corresponding  $R_{scia2gome}$  measurements using Akima interpolation scheme. The ratio of  $R_{scia2gome}$  to the  $R_g$  is calculated based on the Equation (4):

$$190 \quad Ratio(\lambda_s) = \frac{R_{scia2gome}}{R_q} \tag{4}$$

The *Ratio* was computed for each GOME pixel overlapped with SCIAMACHY pixels as a function of wavelength for each site and the three spectral windows selected by FDR4ATMOS. The spectral windows are shown in Table 2.

**Table 2.** Spectral windows selected for FDR4ATMOS and used for harmonization. For the NIR, the  $O_2$  A-band is excluded for the TFs FDR4ATMOS manual (2024).

| Spectral window | GOME-1 | SCIAMACHY | Wavelength interval(s) |  |  |  |
|-----------------|--------|-----------|------------------------|--|--|--|
|                 | band   | band(s)   | (nm)                   |  |  |  |
| UV              | 2B     | 09 & 10   | 313–347                |  |  |  |
| VIS             | 3      | 15        | 424–495                |  |  |  |
| NIR             | 4      | 26        | 756–757 & 773–774      |  |  |  |

# 4.1.4 Removal outliers

The interquartile range (IQR) is a robust measure of the statistical dispersion of the samples. IQR includes the data between  $Q_1$  (25% of data) and  $Q_3$  (75% of data). IQR was applied to reflectance ratios from Equation (4) to remove the outliers and focus on the central portion of the data.

$$IQR = Q_3 - Q_1 \tag{5}$$

The ratios, from Equation (4), falling below  $Q_1 - 1.5 \cdot IQR$  or above  $Q_3 + 1.5 \cdot IQR$  were considered potential outliers. The factor 1.5 was chosen based on the assumption that the ratios had a normal distribution. It was further validated through trial and error to ensure that with the chosen factor, approximately 99% of the ratios remained within these bounds.

Outliers 
$$< Q_1 - 1.5 \cdot IQR$$
 or Outliers  $> Q_3 + 1.5 \cdot IQR$  (6)

## 4.1.5 Transfer functions

205

The TFs represented the ratios of Equation 4 as a function of wavelength. For the UV and VIS spectral windows, the TF were modeled based on a weighted least squares fit of a third-degree polynomial function after removing outliers based on IQR. Only data with a CF less than 0.25 were considered.

For the NIR spectral window, 1 nm-width wave intervals to the left and right of the  $O_2$  A-band were considered for the TFs. The median of these average ratio measurements was taken as the wavelength-independent TF to be applied to GOME

225

reflectance. The  $O_2$  A-band was excluded from the harmonization process, as it is an observational window with high atmospheric sensitivity, leading to strong variability.

## 210 4.2 PMD analysis and filter for improvement of the cross-calibration

## 4.2.1 PMD-based analysis on pixel homogeneity

GOME and SCIAMACHY spectrometers were sensitive to polarization. Atmospheric polarisation could be retrieved from PMD measurements. For our purpose, we utilize the fact that the PMD detectors operate at a significantly higher temporal resolution than the main spectral channels. For GOME, the readout rate was 16 times faster than the main channels, whereas the smallest SCIAMACHY pixels had 6 PMD readouts. The structure and sensitivity of the PMD detectors make their data particularly valuable for assessing pixel homogeneity in the cross-calibration study. In this work, PMD channels spanning the UV (PMD-1), VIS (PMD-2), and NIR (PMD-3) ranges from both GOME-1 and SCIAMACHY were analyzed for collocated observations.

The comparison was performed by calculating the statistical indicators that describe the correlation and the variation among PMDs of overlapping pixels. The statistical indicators include the average  $(\mu)$ , the standard deviation  $(\sigma)$ , the Pearson correlation coefficient  $(R^2)$ , and the coefficient of variation (CV).

High standard deviation in PMD values within GOME pixels, as well as within SCIAMACHY pixels that intersect the same GOME pixel, indicates a low level of spatial homogeneity in the scan direction. Additionally, a large difference between  $\sigma$  and CV of PMDs from GOME and the corresponding intersecting SCIAMACHY pixels suggests the presence of non-homogeneous conditions in the cross-scan direction, or of temporal variability.

The following steps were considered to perform the PMD analysis:

- 1. The coordinates of the overlapping GOME and SCIAMACHY pixels of the three PICS were retrieved. As an example, Fig. 3 illustrates the intersections for two GOME pixels with 24 SCIAMACHY pixels. A polygon is created for each intersection.
- 2. The created polygons were used to filter the spatial coordinates (longitude and latitude) of the PMD measurements from the corresponding GOME and SCIAMACHY PMD channels, ensuring that only PMD measurements intersecting the overlapped GOME and SCIAMACHY pixels were included.
  - 3. For each GOME pixel used in the cross-calibration,  $\mu$ ,  $\sigma$ , and CV of the PMD measurements from both GOME and the overlapped SCIAMACHY pixels were computed.

$$\sigma_{\text{PMD}_{\text{sensor},j}} = \sqrt{\frac{1}{n} \sum_{k=1}^{n_j} (\text{PMD}_{\text{sensor},j,k} - \mu PMD_{sensor,j})^2}$$
(7)

where sensor refers to either GOME or SCIAMACHY, n is the number of measurements from PMDs,  $\mu_{PMD_{sensor,j}}$  is the average of PMD measurements for an overlap, j is the PMD channel, and k is the index over PMD measurements in the exact overlap (not in the whole pixel). The CV (%) is computed based on the ratio of  $\sigma$  and  $\mu$ 

$$CV = \frac{\sigma_{\text{PMD}_{\text{sensor},j}}}{\mu_{\text{PMD}_{\text{sensor},j}}} \times 100\%$$
 (8)

#### 240 4.2.2 PMD filter

245

The spatial variability for each collocated GOME pixel with SCIAMACHY pixels was quantified by estimating the standard deviations of the PMD measurements within the intersected areas. The PMD-based filter was designed using the absolute difference between the standard deviations of GOME and SCIAMACHY reflectance (D), as shown in Equation (9). A percentile-based threshold was then defined to identify GOME pixels with low variability, using the 25th percentile of the absolute differences. This 25th percentile threshold represents the minimum detectable variability in the PMD measurements.

$$D = |\sigma_{\text{PMD}_{\text{GOME},j}} - \sigma_{\text{PMD}_{\text{SCIAMACHY},j}}|$$

$$\tag{9}$$

where D is the absolute difference, and j refers to the channel index.

$$T = P_{25}(D) \tag{10}$$

where  $P_{25}$  means the value below which 25% of the absolute differences lie.

This filtering was applied to ensure that only homogeneous pixels—those with minimal variability—contributed to the derivation of a new version of the TFs, referred to as TFs with PMD. These were then compared to the originally derived functions, referred to as TFs without PMD, to assess the impact of filtering on the TFs.

#### 5 Results

# 5.1 Statistics and geographic distribution of GOME pixels used in derivation of TFs

A total of 151 GOME pixels with CF less than 0.25 were found to overlap with SCIAMACHY pixels in scanlines that include the PICS and their surrounding area (see Fig. 4). Each GOME scanline consists of three ground pixels (east, nadir, and west, according to VZA). In each scanline, only one ground pixel can intersect the PICS. Therefore, approximately 34% of these GOME pixels intersect the PICS. More measurements can contribute to the derivation of TFs when all scanlines are considered.

Table 3 summarizes the distribution of used GOME pixels based on VZA, along with the number of pixels intersecting the PICS. According to these statistics, GOME pixels with west VZA were found to have the highest intersection rate, and lower for east and nadir. The geographic distribution of the GOME pixels used in this study is illustrated in Fig. 4, showing that most of them are located over desert surfaces, except some east-VZA pixels in Arabia2 and Sudan1 that crossed water surfaces, either from the ocean or the river.

Figure 4. GOME pixels distribution over PICS and surrounding regions. (a) Sudan1, (b) Arabia2, and (c) Sudan1. The GOME pixels are plotted according to VZA, whereas east, nadir, and west VZAs are represented by orange, green, and blue, respectively. PICS with dimensions  $100 \text{ km} \times 100 \text{ km}$  are shown as a black dashed square.

**Table 3.** Statistics of GOME pixels used in the derivation of TFs based on VZA for each PICS, and the number of GOME pixels that intersected with PICS.

| PICS                               | Sudan1 |       | Arabia2 |      | Libya4 |      |      |       |      |
|------------------------------------|--------|-------|---------|------|--------|------|------|-------|------|
| VZA class                          | West   | Nadir | East    | West | Nadir  | East | West | Nadir | East |
| Number of pixels                   | 10     | 10    | 10      | 22   | 23     | 22   | 15   | 15    | 24   |
| Number of pixels intersecting PICS | 0      | 9     | 1       | 13   | 3      | 7    | 7    | 1     | 10   |

## **5.2** Transfer functions

The ratios from Equation (4) were computed for overlapping GOME and SCIAMACHY pixels using two approaches: (i) considering only pixels that directly intersect the PICS, and (ii) including all ground pixels in a scan that had an overlap with the PICS. The comparison revealed minimal differences between the two approaches. Fig. 5 presents the median values and corresponding standard deviations for Bands 2B (Fig. 5a) and 3 (Fig. 5b). The largest discrepancy was observed for Band 2B in the east viewing geometry, with a maximum difference of approximately 1.5%. Overall, the magnitude of uncertainties was found to be consistent across the different bands.

**Figure 5.** Median and the standard deviation values of ratios of Equation (4) for the only ground pixels that include the exact PICS area (black color and labelled "Intersected") and all collocated pixels over the PICS and surrounding areas (colors and labelled "All") for (a) Band 2B west, nadir, and east VZA (b) Band 3

We chose to use ratios from all colocated pixels to increase the statistics. The resulting ratios are illustrated in Figure 6. In general, although the ratio patterns varied across the bands, each band exhibited a consistent pattern across the PICS. For band

2B (Figure 6(a-c)), the ratios—particularly at the shorter wavelengths—were dependent on the VZA. The ratios were higher for the overlapping pixels with eastward VZA compared to those with nadir and westward VZA.

For band 3 (Figure 6(d-f)), similar to band 2B, the ratios increased by wavelength, yielding a positive slope indicating a across the PICS. For band 4 (Figure 6(g-i)), the  $O_2$  A-band was excluded from the calculation of the TFs. The ratios on both sides of the  $O_2$  A-band were relatively constant and near 0.95.

The derived TFs for each spectral band, median, and standard deviation value are shown in Fig. 7. For Bands 2B and 3, the TFs were modeled using third-degree polynomial fits, while for Band 4, a nearly constant value of 0.94 was used. The uncertainty in the TFs is represented by error bars corresponding to one  $(\sigma)$ .

Among the VZA classes in Band 2B, the nadir view exhibited the lowest uncertainty. In contrast, the uncertainties for the west and east VZA classes were, on average, approximately 3 and 2 times higher than that of the nadir VZA, respectively. For Bands 3 and 4, larger uncertainties were observed. On average, the uncertainties in Bands 3 and 4 were about 4 times greater than the uncertainties in the Band 2B nadir TFs.

Figure 6. Ratios  $(R_{SCIA2GOME}/R_g)$  of each band (rows) and selected PICS (columns). For band 2B (a-c), the ratios are classified based on VZA to the west (green), east (blue), and nadir (red). (d-f) and (g-i) refer to ratios for band 3 and band 4, respectively. Each line curve represents the ratios of one GOME pixel overlapped with several SCIAMACHY pixels. These ratios were accompanied by cloud fractions less than 0.25.

Figure 7. The transfer functions are derived by fitting the median values of the spectral channels' ratios (dots) using a third-degree polynomial function (curves) for (a) band 2B, and (b) band 3. (c) A constant function (orange line) of approximately 0.95 was used for band 4, excluding the  $O_2$  A-band. The error bars indicate one standard deviation and are shown every 5 steps for Bands 2B and 3, and 1 step for Band 4.

## 5.3 PMD analysis of both sensor pixels used in the cross-calibration

The PMD analysis of the GOME and SCIAMACHY pixels used in the derivation of TFs, based on PMD channels 1, 2, and 3—representing Bands 2B, 3, and 4, respectively—is shown in Fig. 8. The analysis includes the statistical indicators  $\mu$ ,  $\sigma$ , and CV.

For Band 2B (Fig. (8a), based on  $\mu$  measurements from PMD channel 1, the PMD values from GOME and SCIAMACHY showed strong correlations, with  $R^2$  values of 0.90, 0.96, and 0.97 for west, nadir, and east VZA pixels, respectively. The PMD measurements exhibited a clear dependence on the VZA. SCIAMACHY tended to underestimate the PMD values compared to GOME for the nadir and east VZA pixels, while a slight overestimation was observed for the west VZA pixels.

For Bands 3 and 4 (Fig. 8(b-c)), the correlation between PMD measurements from GOME and SCIAMACHY was lower, with  $R^2$  values of 0.71 and 0.79, respectively, indicating weaker agreement compared to Band 2B. In both cases, SCIAMACHY generally underestimated the PMD values relative to GOME.

The  $\sigma$  values of PMD measurements for Band 2B from both GOME and SCIAMACHY (Fig. 8d) were relatively low, indicating more spatially uniform and homogeneous pixels. In contrast, for Bands 3 (Fig. 8e) and 4 (Fig. 8f), the  $\sigma$  values of GOME PMDs were higher than those of SCIAMACHY, suggesting that less homogeneous pixels were involved in the cross-calibration for these bands. A similar trend was observed in CV: lower variability was seen in Band 2B (Fig. 8g) compared to Bands 3 (Fig. 8h) and 4 (Fig. 8i). Furthermore, the PMD measurements of GOME pixels exhibited higher variability than those of SCIAMACHY.

Figure 8. PMD-based analysis of GOME and SCIAMACHY pixels used in the derivation of TFs. Subplots (a–c) show the mean ( $\mu$ ) PMD values for channels 1, 2, and 3, corresponding to Bands 2B, 3, and 4, respectively. Subplots (d–f) and (g–i) present  $\sigma$  and CV for the same channels, respectively, providing additional insights into the variability and homogeneity of the observed pixels.

# 5.4 PMD-based filter

The PMD-based filter was applied to the GOME pixels used in the derivation of the TFs of Fig. 7. This filter was applied separately for each site and each PMD channel, as illustrated in Fig. (9). The red scatter points represent the GOME pixels for which the absolute relative difference in the PMD standard deviation compared with SCIAMACHY pixels was less than 25%. The filter resulted in a significant reduction in the sample size. After applying this filter, 39 out of 151 GOME pixels were retained for the cross-calibration, yielding the TFs denoted as TFs with PMD.

**Figure 9.**  $\sigma$  PMD GOME vs.  $\sigma$  PMD SCIA of the PICS (Sudan1, Arabia2, and Libya4). Each site includes three subplots: (left) PMD1, (middle) PMD2, and (right) PMD3. The red scatter points represent the GOME pixel that was used in the derivation of TFs with PMD after fulfilling the criteria in Equation (10).

## 5.5 Transfer function with PMD filter

The fitted TFs for the bands without applying the PMD filter (black dashed curves) were consistent with the TFs derived when the PMD-based filter was used (colored curves), as shown in Figs. (10(a–c)). The differences in the median values of both TFs are highlighted in Figs. (10)(d–f). The maximum absolute difference was 1.75% for Band 2B west VZA (Fig. 10(d)). Overall, the TF values (with and without the PMD-based filter) changed only slightly and not significantly. However, the uncertainties of the TFs were markedly reduced when the PMD-based filter was applied (Fig. (10) (g-i). For Band 2B west, the uncertainty decreased by up to 70% (Fig. 10(g)). In contrast, there was no uncertainty reduction for Band 2B nadir and east VZA. For Bands 3 and 4, the uncertainty reduction reached up to 5.5% (Fig. 10(h))and 22% (Fig. 10(i)), respectively.

**Figure 10.** The comparison of TFs with PMD (solid lines) and without PMD (dashed lines) (a) Band 2B west, nadir, and east VZA, (b) Band 3, (c) Band 4. The reduction in the uncertainty for (d) Band 2B west, nadir, and east VZA, (e) Band 3, and (f) for Band 4.

#### 6 Discussion

325

335

340

## 6.1 Challenges on spectrometer cross-calibration and mitigation strategies

The main challenges mentioned in the introduction about the GOME and SCIAMACHY cross-calibration were tackled in the methods by applying the following strategies:

First, the cross-calibration was performed using observations acquired over PICS, which were characterized by highly stable atmospheric and meteorological conditions and community-accepted calibration sites. The probability of rapid changes in atmospheric and meteorological conditions within the 30-minute acquisition time difference between sensors is relatively low for these sites.

Second, the observations were filtered based on CF to ensure that only data collected under nearly clear-sky conditions were used. This filtering step was essential to ensure consistency across the data sets involved in the cross-calibration.

Third, the PMD-based analysis approach was applied as an indicator of pixel homogeneity in the cross-calibration. This approach helped exclude non-homogeneous pixels and ensured more uniform pixel textures, thereby reducing the uncertainty of TF. This was possible thanks to the higher sampling frequency of PMD measurements compared to the main spectral channels.

Finally, spectral regions with strong absorption features, such as the  $O_2$  A-band, were excluded from the cross-calibration process to avoid the influence of atmospheric absorption variability.

## 6.2 Transfer functions from PICS

Including all measurements from the complete scanline, rather than limiting the analysis to only those directly intersecting with the PICS, did not significantly impact the cross-calibration results. This is likely due to the similar scene characteristics and surface reflectance signals typical of desert environments. The advantage of PICS was not significantly lost for the spectrometer ground pixel that extends beyond the defined boundary of PICS. Additionally, the small time difference in the collocation helps to mitigate significant differences in the content of the cross-calibrated pixels.

The ratios of reflectance values of both sensors were in the range between 0.8 and 1.1 for the GOME bands before applying the PMD-based filter (see Fig. 4). The median values, aiming to minimize the impact of outliers on the derived TFs, of the ratios were used in fitting third-degree polynomial functions. GOME data were corrected for degradation, but for only one scan viewing angle, where the instrument was looking at the sun. However, the derived TFs of band 2B exhibited a stronger dependence on VZA compared to those for bands 3 and 4. This behavior is attributed to the significant UV degradation of the GOME instrument since 2001, which was scan mirror-dependent and varied with the viewing angles (Tilstra and Stammes (2006)), particularly at shorter wavelengths. In contrast, the degradation was less pronounced at longer wavelengths, resulting in reduced VZA dependency for bands 3 and 4. Consequently, the impact of degradation on TFs is expected to be more substantial over time. A notable observation is that the uncertainty appears to be wavelength-dependent. The uncertainties in Bands 3 and 4 were higher compared to Band 2B. This may be attributed to reduced scene homogeneity at longer wavelengths. Further details on this relationship are provided in Section (6.4).

360

## 6.3 PMD measurements on describing the homogeneity of scenes

GOME and SCIAMACHY were both equipped with PMDs, which provide sub-pixel-scale measurements. The PMD measurements from GOME pixels involved in the cross-calibration were consistently higher than those from the corresponding SCIAMACHY pixels due to: (i) GOME PMDs had roughly double the spectral bandwidth compared to SCIAMACHY, and the optics are different, (ii) the GOME measurements at the equatorial crossing time of 10:30 against SCIAMACHY at 10:00, hence, the GOME PMDs would receive larger signals, (iii) PMD data on Level 1b were not corrected with the cos (phi) factor (as the reflectance was), (iv) the larger values in PMDs GOME could explain that we have more sampling and less averaging than for a SCIAMACHY pixel. Nevertheless, different levels of correlation were observed between the two instruments, with the strongest correlation for Band 2B.

The PMD measurements from both GOME and SCIAMACHY exhibited low  $\sigma$ , indicating that the selected pixels were generally spatially homogeneous. However, the analysis revealed that the degree of spatial uniformity was wavelength-dependent. A broader range of  $\sigma$  and CV values was found for Bands 3 and 4 compared to Band 2B, suggesting reduced pixel homogeneity at longer wavelengths. This reduction in homogeneity contributed to higher uncertainties in the TFs derived for Bands 3 and 4.

# 6.4 PMD-based filter on the improvement of the cross-calibration

The PMD analysis highlights the crucial role of PMD measurements in assessing pixel homogeneity for cross-calibration. It provides a basis for developing robust filtering criteria using PMD-derived indicators, which enable the exclusion of spatially non-uniform pixels. Overall, most cross-calibrated pixels from both sensors were spatially homogeneous, with absolute differences in standard deviation (σ) below 0.01. Applying the PMD-based filter led to the exclusion of a considerable number of GOME pixels from the TF derivation. Nevertheless, the absolute TF values with and without the filter showed no significant differences, while the uncertainties of many TFs were substantially reduced. This selective adjustment demonstrates that the PMD-based filter effectively improves TF estimates in the presence of non-homogeneous pixels. Ultimately, this approach broadens the applicability of TF derivation by enabling the inclusion of a more diverse range of surface types and reflectance conditions, rather than relying solely on desert scenes.

## 6.5 Limitation of the derived TFs

The derived TFs still exhibit some limitations and constraints that should be addressed to improve their accuracy and representativeness further:

(i) Temporal dependency: The derived TFs depend on data from the year 2003. It has also been reported by Owda and Lichtenberg (2025, submitted) that degradation was severe toward the end of the GOME life mission. Since the TFs are expected to be influenced by degradation, especially in the UV, it is recommended to apply the reflectance degradation to the GOME data before deriving TFs to ensure their long-term consistency. This is envisaged for the next version of TFs of the FDR4ATMOS project.

(ii) Scene dependency: The TFs were derived using scenes with similar surface characteristics, primarily desert regions. As a result, the derived TFs are primarily tailored to a specific type of reflectance signal and may not generalize well to other surface types. To enhance their representativeness, it may be advisable to include additional reflectance signals from more diverse surfaces, such as those in oceanic and polar regions, provided that the selected scenes meet established homogeneity criteria. That work is envisaged for a follow-up paper.

#### 7 Conclusion

GOME and SCIAMACHY were well-known passive spectrometers with overlapping measurements of 10 years. Data from both spectrometers spanning two decades are available. To fully leverage data from heritage missions like GOME, it is important to harmonise their measurements based on reference observations. Several studies reported on the stability of SCIAMACHY during its life mission and the better degradation model compared with GOME. That makes the SCIAMACHY a reliable reference for the cross-calibration.

The cross-calibration of spectrometers like GOME and SCIAMACHY is more complicated than the cross-calibration of imagers. The cross-calibration of GOME and SCIAMACHY presented unique challenges. There was no spatial collocation, and the sensors had different spatial resolutions. Furthermore, spectral structures had to be kept for the retrieval of trace gases. Strategies were introduced to address these challenges before the derivation of the TFs. These strategies shared the common goal of standardizing the conditions of overlapping pixels by reducing the influence of atmospheric and meteorological effects. Therefore, only measurements with a CF of less than 0.25 were used in the analysis. Furthermore, the spectral channels with strong absorption features were excluded from the derivation of TFs.

The ratio of reflectance values for the overlapping pixels was consistent across spectral channels and PICS. The TFs for GOME bands 2B and 3 were three-degree polynomial functions, and for band 4, a constant value near 0.94. The derived TFs for the short wavelengths of GOME band 2B were more dependent on VZA compared with longer wavelengths. It found that the uncertainty in TFs for Band 2B nadir was the lowest among the other TFs.

The paper presented an approach for assessing scene homogeneity using PMD measurements. These measurements were characterized by higher spatial resolution, down to the sub-pixel level, and high sensitivity, and were effective in capturing the spatial variations within the observed scenes. The PMD analysis demonstrated a correlation between measurements of PMDs of GOME and SCIAMACHY. The homogeneity of pixels was wavelength-dependent. The spatial uniformity of overlapped pixels of Band 2B was higher than Bands 3 and 4, which impacts the uncertainty of TFs among the bands.

The PMD-based analysis enables the identification of non-homogeneous overlapping pixels, allowing them to be excluded before the derivation of TFs. Subsequently, the ranges of ratios used to derive TFs were significantly reduced, leading to a higher level of convergence for fitting the polynomial functions. This improves the overall reliability of the cross-calibration process. The PMD-based filter adjusts the TFs only when necessary—specifically when non-homogeneous pixels are present—resulting in reduced uncertainties. This approach highlights the potential for expanding the dataset to include a larger number of observations from non-PICS sites by enabling the identification of pixels that meet homogeneity criteria.

420

A forthcoming paper will explore the potential of utilizing scenes characterized by a variety of surface textures and reflected signal properties. By extending the analysis beyond stable PICS to include more diverse land cover types and reflectance characteristics. The outcomes are expected to provide valuable insights into optimizing pixel selection criteria and enhancing the reliability of satellite sensor calibration in more complex and variable scenes.

# 8 Data availability and tools

- SCIAMACHY Level-1b data: https://doi.org/10.5270/EN1-5eab12a
- SCIAMACHY 1c Tool: https://earth.esa.int/eogateway/tools/scial1c-command-line-tool
  - GOME-1 Level-1b data: https://earth.esa.int/eogateway/instruments/gome/products-information
  - FDR4ATMOS products: https://doi.org/10.5270/ESA-852456e

Author contributions. AO: Conceptualization, Methodology, Programming, Investigation, Writing—original draft. MCE: Programming, Writing—review & editing. SS: Conceptualization, Methodology, Writing—review & editing. GL: Conceptualization, Methodology, Writing—review & editing. BA: Programming.

Competing interests. The authors declare no competing financial or personal interests that could have influenced this work.

Acknowledgements. The FDR4ATMOS project, initiated by ESA, provides long-term records of Earth observation Level 1 parameters (radiance, irradiance, reflectance) to improve the performance of mission datasets (FDR4ATMOS website).

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
