# Peer review of "Cross-calibration of GOME and SCIAMACHY Spectrometers Enhanced by Polarization Monitoring Devices Data"

_EGUsphere, 2025_

## Referee Comment (RC1)

**Review of "Cross-calibration of GOME and SCIAMACHY Spectrometers Enhanced by Polarization Monitoring Devices Data"**

Comments based on https://egusphere.copernicus.org/preprints/2025/egusphere-2025-4942/ Preprint, retrieved 18 November 2025

**General comments**

Dear authors,

The paper is in general written in a clear and succinct manner and the reasoning is good to follow. The subject is highly relevant for long term atmospheric data records and this pre-study shows how the PMDs could be used to improve the cross calibration between GOME and SCIAMACHY.

However there are some points where the paper could be improved: Table 1 does not list the differences between GOME and SCIAMACHY, this is probably not intentional and should be corrected. Furthermore some of the information on the instruments should be given earlier in the text and not only in the discussion.

The fact that the PMDs can be used to extend the cross calibration beyond the PICS can be emphasized even more in the abstract. If the paper is framed as a pre-study for future work, the low number of pixels used to derive the transfer functions is explainable.

You have chosen a structure where the paper is divided in methods, results and discussion. It might improve the overall readability of the paper and avoid lengthy repetitions if you outline the different steps in a few bullets and discuss each step in detail including the used method, result and discussion. And of course summarize in the conclusion.

Please find below specific comments on the content and in a separate table technical comments on typos, phrasing and layout of figures.

**Specific comments**

| Item | Section | Line | Comment |
|---|---|---|---|
| SC1 | 0 | 14 | Please add "from GOME and SCIAMACHY". It's otherwise not clear that both instruments had PMDs on board. |
| SC2 | 1 | 60 | "is has been assumed", can this point not be proven or confirmed? I would rephrase and include a reference where it has been shown. |
| SC4 | 2 | 103 | "four channels" Which ones? You only name 3 (UV, VIS, NIR) |
| SC5 | 2 | 104 | Spectral resolution in NIR is missing. |
| SC3 | 2 | 115 | Table 1 and it's caption is identical to Table 2. The listed differences between the instruments should include the LTAN, size and spectral resolution of the PMDs, the spectral resolution and range of both instruments, spatial resolution,... |

| | | | |
|---|---|---|---|
| SC6 | 2 | 119 | You refer that you use SCIAMACHY L1c data, but then you mention the preprocessing from L1b to L1c, please correct. Either L1c is input and you can discard the discussion on the cluster concept and refer instead to the specification of the tool or you start from L1b data and then do the pre-processing. |
| SC7 | 2 | 133 | Please explain why you use reflectance (most instrument effects divide out) |
| SC8 | 2 | 140 | What is the spatial resolution of the PMDs? Please be specific. |
| SC9 | 4 | 188 | What is the Akima interpolation scheme? Please explain or give a reference. |
| SC11 | 4 | 204 | Why do you use a 3rd degree polynomial? Have you done any over- and underfitting tests? The UV and VIS still show some features which are not fitted but consistent for all cases, what is the reasoning to not fit them? Please explain this in the paper. (I assume you will to keep spectral features) |
| SC12 | 4 | 213 | higher temporal resolution: the specifics could be added to the updated Table 1 and referred to here. |
| SC13 | 4 | 244 | threshold $T$ (T is not defined in Eq.(10)) |
| SC14 | 4 | Table 1 | You could add in the caption 757-773nm for the O2 A-band |
| SC15 | 5 | 255 | The number of 151 is quite low, this is for one year of data? It might be good to repeat that here. |
| SC16 | 5 | 258 | "all scanlines" what do you mean by all scanlines? Over different regions? Far away from the PICS. Or do you mean the groundpixels close to but not overlapping with the PICS? |
| SC17 | 5 | 268 | Are the differences between Nadir, East and West not present for VIS and NIR? Please add this in the text or show the different directions for VIS and NIR in a plot. |
| SC18 | 5 | 279 | "nearly constant value", please be more specific, the fit is close to 0.94 and you then chose this value? Or did you average the value left & right of the O2 A-band? |
| SC19 | 5 | 283 | Why are the uncertainties larger? Is that because you combined all views? Would it improve if you distinguish East, West and Nadir? |
| SC20 | 5 | 286 | Some more instrument details would be useful (add in Section 2). Are the PMD matched to the spectral channels in wavelength? Earlier in the text you write they are broadband. Is the field of view the same as for the spectrometer views? |
| SC21 | 5 | 300 | PMD variability GOME & SCIAMACHY: do you have an explanation why the variability is higher? Are they bigger (please add PMD size to Section 2) ? |
| SC22 | 5 | 306 | Only 39 pixels left over? The seems rather little. |
| SC23 | 5 | Fig. 4 | (c) Sudan 1 -> Libya 4? |

| | | | |
|---|---|---|---|
| SC24 | 6 | 328 | "higher spatial sampling frequency" , how much higher the sampling is should be included in Section 2 |
| SC25 | 6 | 347 | This may be attributed… Please be specific, is it a guess or do you have any evidence for this statement? |
| SC26 | 6 | 350 | It is a bit late to provide all this PMD and instrument information only in the discussion. Please move all this information to Section 2. |
| SC27 | 6 | 358-362 | This part has already been discussed in the results section. It does not need to be repeated here. And as in line 347, can you show that the sites are less homogenous in these wavelengths than in others (maybe from other instruments?) |
| SC28 | 6 | all | For the discussion, apart from extending the scenes to derive TFs, what is the wider implication can this method also be used to cross-calibrate other instruments? Could this also be used to cross-calibrate to instruments without PMD but also a higher spatial sampling for certain wavelengths (small pixel columns in OMI for example). It is a good point to close the loop with the abstract where the need to cross calibrate instruments over many decades is described. |
| SC29 | 7 | 397 | "spectral channels with strong absorption" : this suggests more than the O2 A-band was excluded, is this correct? Then this should be mentioned already much earlier. If not just refer to the O2 A-band. |
| SC30 | 7 | 417 | Please call it cross calibration, the absolute calibration of both sensors has not been investigated. |

**Technical comments/typos**

For all figures, please check if these can be read by colourblind people. Make use of different linetypes and use a different colour scheme.

| Item | Section | Line | Comment |
|---|---|---|---|
| TC1 | 0 | 5 | was --> is |
| TC2 | 0 | 14 | presented-->presents |
| TC3 | 1 | 30 | The focus arises -> this sentence does not read well, please rephrase |
| TC4 | 1 | 47 | is one of the well-known methods -> is a well-known method, the "one of" implies that you should also mention other methods |
| TC5 | 1 | 49 | ". Many other man-made" implies that deserts or oceans are man-made. Rephrase to ", or man-made features, …" |
| TC6 | 2 | Fig 1 | Please increase the size of axis and text labels. |
| TC7 | 3 | Fig 2 | Caption: "except" -> excluding. Figure: please make sure the steps in the figure have the same name as in the text. |
| TC8 | 4 | Fig 3 | Please increase the size of axis and legend and make it colourblind proof. The degree sign is missing for the numbers |

| | | | |
|---|---|---|---|
| TC9 | 4 | 175 | is *the* pseudo |
| TC10 | 4 | 185 | The sentence reads difficult, there seems to be a word missing. Consider to rephrase: "C1 and C2 are the correction factor and offset for scene dependent effects" |
| TC11 | 4 | 203 | represented ->represent |
| TC12 | 4 | 206 | 1nm-width wave -> 1nm wide wavelength intervals |
| TC13 | 5 | 276 | indicating a ____ across (word missing). |
| TC14 | 5 | Fig. 6-10 | Figures 6, 8, 9 and 10 should have less white around them and the text should be larger. If you change the aspect ratio and scale to the actual data points, it would be easier to read. |
| TC15 | 5 | 315 | ))_ and (space missing) |
| TC16 | 5 | Fig 10 | Caption: panels g to i are not described |
| TC17 | 6 | 328 | of *the* TF |
| TC18 | All | All | Please reconsider when you use the past tense, for example cross calibration *is (not was)* important. |
| TC19 | | Fig 6/10 | When plotting the NIR data vs wavelength, please consider a interrupted scale and focus the plot more on where you do use data. |

---

## Author Comment (AC1)

Referee comment on "**Cross-calibration of GOME and SCIAMACHY Spectrometers Enhanced by Polarization Monitoring Devices Data**" by Owda, A., Coldewey-Egbers, M., Slijkhuis, S., Lichtenberg, G., and Aberle, B., https://doi.org/10.5194/egusphere-2025-4942, 2025.

**We are grateful to the anonymous referee for taking the time to review our manuscript and provide us with constructive comments. We have carefully considered all comments and suggestions and thoroughly revised the manuscript.**

**In this response letter, we intend to address all comments, concerns, and suggestions point by point. The response will be highlighted in blue and the corresponding modification in the revised manuscript in orange. The original comment from the anonymous referee will be in black.**

**General comments:**

*"The paper is in general written in a clear and succinct manner and the reasoning is good to follow. The subject is highly relevant for long term atmospheric data records and this pre-study shows how the PMDs could be used to improve the cross-calibration between GOME and SCIAMACHY. However there are some points where the paper could be improved: Table 1 does not list the differences between GOME and SCIAMACHY; this is probably not intentional and should be corrected. Furthermore some of the information on the instruments should be given earlier in the text and not only in the discussion. The fact that the PMDs can be used to extend the cross calibration beyond the PICS can be emphasized even more in the abstract. If the paper is framed as a pre-study for future work, the low number of pixels used to derive the transfer functions is explainable. You have chosen a structure where the paper is divided in methods, results and discussion. It might improve the overall readability of the paper and avoid lengthy repetitions if you outline the different steps in a few bullets and discuss each step in detail including the used method, result and discussion. And of course summarize in the conclusion"*

**Response: Thank you very much for the general comments and the positive impression about our paper.**

Regarding Table 1, it was not intentionally duplicated with Table 2. Table 1, which compares the differences between GOME and SCIAMACHY, has been added to the revised manuscript.  [Fixed]

Additionally, we agree that further details on the instruments, particularly the PMDs, should be addressed in the beginning. The changes to the manuscript regarding these comments will be addressed within the specific comments section.  [Fixed]

We agreed that further emphasis should be placed on extending the cross calibration beyond the PICS; therefore, the abstract was modified.  [Fixed]

Lines 9-10 (Abstract)

"Sub-pixel variability analysis that allows us to evaluate the homogeneity and similarity of the scene as observed by GOME-1 and SCIAMACHY and thus reduce the uncertainties in the cross-calibration process."

Lines 25-27 (Abstract)

"This work highlights the potential for extending the cross-calibration beyond traditional PICS and ideally suited scenes, increasing robustness across varied surfaces and atmospheric conditions."

We stick with our chosen structure of the research paper. We have removed many repetitions. For some sections, such as 2.3, 4.2.1, and 6.1, we outlined the discussion in bullets.

**Specific comments**

Response: Thank you very much for the specific comments. We believe that your comments were very useful, and fixing these comments helps to improve the readability of the paper.

We have dealt with your comments one by one as follows:

SC1 0 14 Please add "from GOME and SCIAMACHY." It's otherwise not clear that both instruments had PMDs on board.

[Figure]
 [Fixed]

In line 16 in the abstract, "... from both GOME and SCIAMACHY …."
* * *
SC2 1 60 "is has been assumed", can this point not be proven or confirmed? I would rephrase and include a reference where it has been shown.

[Fixed]

The sentence has been paraphrased, and the same reference is still there.

Line 63: Many studies assumed that the PICS remain temporally invariant
* * *
SC4 2 103 "four channels"—which ones? You only name 3 (UV, VIS, NIR)

[Fixed]

The sentence has been modified and clarified. The four refers to the number of detectors the GOME had, which is 4.

Line 108: "... and four detectors (1A, 1BA, 3, and 4)"
* * *
SC5 2 104 Spectral resolution in NIR is missing.

[Fixed]

The spectral resolution of NIR is added

**Line 108:** "... and 0.4 nm in the VIS and NIR ".
* * *
**SC3 2 115 Table 1 and it's caption are identical to Table 2. The listed differences between the instruments should include the LTAN, size and spectral resolution of the PMDs, the spectral resolution and range of both instruments, spatial resolution,…**

[Fixed]

Table 1 and Table 2 are different now. Table 1, which shows the differences between GOME and SCIAMACHY, is provided in the revised manuscript. Table 2 in the original manuscript becomes Table 4 in the revised manuscript.

See Table 1
* * *
**SC6 2 119 You refer that you use SCIAMACHY L1c data, but then you mention the preprocessing from L1b to L1c, please correct. Either L1c is input and you can discard the discussion on the cluster concept and refer instead to the specification of the tool or you start from L1b data and then do the pre-processing.**

[Fixed]

We decided to remove the text regarding L1b SCIAMACHY and the tool we used to convert L1b to L1C. The process will start with the input of L1b GOME and L1c SCIAMACHY.

Line 122: "The fully calibrated Level 1 products from GOME (Level 1b) and SCIAMACHY (Level 1c) are used as input for the cross-calibration method."
* * *
**SC7 2 133 Please explain why you use reflectance (most instrument effects divide out)**

[Fixed]

Further explanation is added for the reason of using the reflectance in:

**Lines 130-133:**

*"Reflectance served as a key parameter for the derivation of the transfer functions (TFs), which were used to update the GOME observations based on SCIAMACHY observations, as defined in Section 4.1.4. Reflectance is used as an instrumental effect common to Earth and Sun observations. Light path, such as etalon interference, is largely canceled throughout the rationing. Furthermore, the retrieval of most atmospheric parameters is based on the reflectance.*

**Lines 170-173:**

*"GOME measured radiance, but it was affected by a randomly varying residual etalon, which is effectively removed when converting to reflectances. Therefore, using reflectances as the basis for the harmonization is advantageous."*
* * *
**SC8 2 140 What is the spatial resolution of the PMDs? Please be specific.**

**[Fixed]**

The spatial resolution of the GOME and SCIAMACHY PMDs are given in Table 1. Further details on PMD from both sensors are provided in the same table as well.

See Table 1
* * *
**SC9 4 188 What is the Akima interpolation scheme? Please explain or give a reference.**

**[Fixed]**

We added the reference for the Akima interpolation.

Line 211. *"....Akima interpolation scheme, which is a method for smoothly interpolating data points using piecewise cubic polynomial (Akima, 1970)."*
* * *
**SC11 4 204 Why do you use a 3rd-degree polynomial? Have you done any over- and underfitting tests? The UV and VIS still show some features which are not**

fitted but consistent for all cases, what is the reasoning to not fitting them? Please explain this in the paper. (I assume you will to keep spectral features)

Yes, we have tested based on which function will preserve the structure of the spectrum and avoid overfitting. It found out that the third-degree polynomial function is more suitable to use.

Yes, we agree that there might be some consistent spectral deviation from a polynomial. However, these remain within the uncertainties. It was decided to use a polynomial on ratios since that is "fitted away" when performing a Differential Optical Absorption Spectroscopy (DOAS) fit to retrieve trace gases. Thus, DOAS fits on the FDR4ATMOS would remain fully consistent with previously obtained and validated trace gases.

See Lines: 226-232

"Among the tested polynomial degrees, a third-degree function was selected, as it adequately represents the spectral behavior while preventing overfitting and suppressing artificial artifacts and spurious high-frequency details. Although small systematic deviations from a purely polynomial behavior may exist, these remain within the measurement uncertainties. The use of a polynomial is justified because broadband spectral structures are removed in the Differential Optical Absorption Spectroscopy (DOAS) retrieval by polynomial fitting. Consequently, this approach ensures that DOAS analyses performed on the FDR4ATMOS data remain fully consistent with previously retrieved and validated trace-gas products."
* * *
SC12 4 213 higher temporal resolution: the specifics could be added to the updated Table 1 and referred to here.

[Fixed]

It was meant to be the frequency of the PMD's measurements, not the temporal resolution. Therefore, the phrase "temporal resolution" is replaced with the frequency of the PMDs. The information about the PMD frequency is given in Table 1.

See Table 1

**SC13 4 244 threshold T (T is not defined in Eq.(10))**

**[Fixed]**

**Equation 10 is updated by adding the threshold term and more explanation in terms of the equation.**

**See Equation 10 and Lines (264-266)**
* * *
**SC14 4 Table 1 You could add in the caption 757-773nm for the O2 A-band**

**[Fixed]**

**Table 1 is now Table 4 after adding some tables based on feedback from other reviewers.  The caption is updated. The caption of Table 4 contains the exclusion wavelength ranges of the $O_2$ A-band.**

**See Table 4**
* * *
**SC15 5 255 The number of 151 is quite low, this is for one year of data? It might be good to repeat that here.**

**[Fixed]**

**Line 194 "... for one year of data"**
* * *
**SC16 5 258 "all scanlines" what do you mean by all scanlines? Over different regions? Far away from the PICS. Or do you mean the groundpixels close to but not overlapping with the PICS?**

**[Fixed]**

**All scanlines mean all ground pixels that can exist with the whole scan, regardless if it intersects with PICS or not.**

**Line 197: "...since some ground pixels are close to the PICS but do not overlap with it"**

**SC17 5 268 Are the differences between Nadir, East and West not present for VIS and NIR? Please add this in the text or show the different directions for VIS and NIR in a plot.**

We have also tested and found there is no differences when we consider VZA for Bands 3 and 4. Therefore, we keep only plots of ratios of Band-2B based on VZA.
* * *
**SC18 5 279 "nearly constant value", please be more specific, the fit is close to 0.94 and you then chose this value? Or did you average the value left & right of the O2 A-band?**

**[Fixed]**

It has been clarified how the transfer function for Band 4 is derived. We have taken the average of the ratios beside the $O_2$ A-band (the left and the right side of the $O_2$ A-band).  The derived TFs for Band 4 (including the $O_2$ A-band) are constant functions with a value of 0.94.

**Line 303-305:**  For Band 4, the TF was obtained by averaging the ratios on the left and right sides of the O2 A-band, yielding a constant value of 0.94, which represents the TF for Band 4, including the O2 A-band (blue dashed line in Fig. 7c)
* * *
**SC19 5 283 Why are the uncertainties larger? Is that because you combined all views? Would it improve if you distinguish East, West and Nadir?**

No, it is not about combining all views. As we mentioned in the previous comment, there is no dependence on VZA for Bands 3 and 4.

In the UV, a significant part of the signal comes from scattering by the atmosphere. The contribution of surface reflection strongly increases with increasing wavelength. Thus, inhomogeneity of the scene due to surface inhomogeneity is expected to increase with wavelength. Since we don't have an exact match of GOME and SCIA ground pixels, surface inhomogeneity may cause a difference in TFs for different satellite overpasses, hence a larger scattering in derived TFs around the "true TFs".

The large uncertainties in Bands 3 and 4 compared with Band 2B are explained in Section 5.3, based on the sub-pixel variability of the pixels used in the cross-calibration in the same section from the PMD analysis (see Fig. 8). The values of standard deviation and coefficient variation are high for Bands 3-4. Therefore, the homogeneity of pixels used in the cross-calibration is less than for Band 2B; this leads to higher uncertainties.

Lines 321-325.

"The $\sigma$ values of PMD measurements for Band 2B from both GOME and SCIAMACHY (Fig. 8d) are relatively low, indicating more spatially uniform and homogeneous pixels. In contrast, for Bands 3 (Fig. 8e) and 4 (Fig. 8f), the $\sigma$ values of GOME PMDs are higher than those of SCIAMACHY, suggesting that less homogeneous pixels are involved in the cross-calibration for these bands. A similar trend is observed in CV : lower variability is seen in Band 2B (Fig. 8g) compared to Bands 3 (Fig. 8h) and 4 (Fig. 8i). Furthermore, the PMD measurements of GOME pixels exhibit higher variability than those of SCIAMACHY."
* * *
SC20 5 286 Some more instrument details would be useful (add in Section 2). Are the PMD matched to the spectral channels in wavelength? Earlier in the text you write they are broadband. Is the field of view the same as for the spectrometer views?

[Fixed]

More instrumental details about PMD are added to the revised manuscript. Additionally, a new table (Table 2) is added, which shows the ranges of the PMD for GOME and SCIAMACHY.

See Table 2 and

Lines 152-154. The wavelength ranges covered by the PMDs are chosen to overlap with the corresponding spectral channels. For instance, the wavelength ranges of the PMD-1 for both GOME and SCIAMACHY (Table 2) align with the wavelength range in the UV band used for the cross-calibration in the FDR4ATMOS (Table 3).
* * *
**SC21 5 300 PMD variability GOME & SCIAMACHY: do you have an explanation why the variability.**

**[Fixed]**

It is expected to have different PMD values due to the the broader range of the PMD wavelength of the GOME compared with SCIAMACHY. We have added Table 2, which gives more information about the wavelength ranges of the PMD channels.

Lines: 152-153 "The wavelength ranges of the PMDs in GOME were broader than those of the corresponding PMDs in SCIAMACHY; hence, different PMD measurement values are expected for SCIAMACHY and GOME."
* * *
**SC22 5 306 Only 39 pixels left over? The seems rather little.**

I agree that 39 pixels are rather few. We believe that 39 GOME pixels overlapped with hundreds of SCIAMACHY pixels can be used to derive the TFs.

Fig. 10 shows the differences in TFs when 151 and 39 pixels were used. It shows that the values of TFs have not changed by more than 1.2%. But the significant changes were for the standard deviation.
* * *
**SC23 5 Fig. 4 (c) Sudan 1 -> Libya 4?**

[Figure]

It has been corrected

Figure 4 caption
* * *
**SC24 6 328 "higher spatial sampling frequency" , how much higher the sampling is should be included in Section 2**

[Figure]

The frequencies of the PMDs are provided in Table 1.

Table 1
* * *
**SC25 6 347 This may be attributed… Please be specific; is it a guess, or do you have any evidence for this statement?**

We have evidence from the PMD analysis performed for GOME and SCIAMACHY. Figures 8(d–i) show that both the standard deviation and the coefficient of variation are more widely spread for GOME than for SCIAMACHY. The scatter points are more dispersed along the x-axis than along the y-axis, indicating that the cross-calibrated pixels are not identical and exhibit a low level of homogeneity.
* * *
**SC26 6 350 It is a bit late to provide all this PMD and instrument information only in the discussion. Please move all this information to Section 2.**

[Fixed]

The information about PMDs are provided in Table 1 and Table 2. Further information about PMDs are provided in the manuscript.

Table 1, Table 2, and lines 144-149.
* * *
**SC27 6 358- 362 This part has already been discussed in the results section. It does not need to be repeated here. And as in line 347, can you show that the sites are less homogenous in these wavelengths than in others (maybe from other instruments?)**

[Fixed]

The repeated part is removed. For the sites are less homogeneous; it was a part of the first paper we submitted to AMT. We have presented a methodology to rank the stability of the PICS based on a group of statistical parameters. Further details on the ranking of the sites can be found in our under-review paper:

https://egusphere.copernicus.org/preprints/2025/egusphere-2025-4639/.
* * *
SC28 6 all For the discussion, apart from extending the scenes to derive TFs, what is the wider implication? Can this method also be used to cross-calibrate other instruments? Could this also be used to cross-calibrate to instruments without PMD but also with a higher spatial sampling for certain wavelengths (small pixel columns in OMI, for example)? It is a good point to close the loop with the abstract where the need to cross calibrate instruments over many decades is described.

[Fixed]

The presented method will be used for cross-calibration of further missions in FDR4ATMOS. We have mentioned in the conclusion the missions we planned to cross-calibrated such as GOME-2A/B/C.

Line 443-445: "The proposed method can be applied in principle whenever higher spatial resolution data are available,
* * *
SC29 7 397 "spectral channels with strong absorption" : this suggests more than the O2 A-band was excluded, is this correct? Then this should be mentioned already much earlier. If not just refer to the O2 Aband.

[Fixed]

We excluded only the $O_2$ A-band from the derivation of TFs.

Caption of Table 4
* * *
SC30 7 417 Please call it cross calibration; the absolute calibration of both sensors has not been investigated.

[Fixed]

We did only cross-calibration and did not perform any absolute calibration investigation.

**Technical comments/typos**

**Response: Thank you very much for the technical comments/typos. We appreciate your language check and apologize for the typos. We have checked the revised manuscript several times to avoid the typos as much as we can.**

**We have dealt with your comments one by one as follows:**

**TC1 0 5 was --> is**

[Figure]
 **[Fixed]**

**Line 5**
* * *
**TC2 0 14 presented-->presents**

[Figure]
 **[Fixed]**

**Line 8**
* * *
**TC3 1 30 The focus arises -> this sentence does not read well, please rephrase**

**[Fixed]**

**The sentence has been paraphrased**

**Lines 34-36: This leads to a focus on the urgent need to address atmospheric change and its impacts**
* * *
**TC4 1 47 is one of the well-known methods -> is a well-known method, the "one of" implies that you should also mention other methods**

[Figure]
 **[Fixed]**

**The sentence has been paraphrased**
* * *
**TC5 1 49 ". Many other man-made" implies that deserts or**

[Figure]

The sentence has been paraphrased
* * *
**TC6 2 Fig 1 Please increase the size of axis and text labels**

[Figure]

Fig 1 has been updated by increasing the size of axis and labels in the revised manuscript.
* * *
**TC7 3 Fig 2 Caption: "except" -> excluding. Figure: please make sure the steps in the figure have the same name as in the text.**

[Fixed]

Fig. 2 has been updated by using the same variables as used in the text.
* * *
**TC8 4 Fig 3 Please increase the size of the axis and legend and make it colorblind-proof. The degree sign is missing for the numbers**

Fig 3 has been updated by increasing the size of the axis and labels and adding a degree sign in the revised manuscript.

[Figure]
* * *
**TC9 4 175 is the pseudo**

**[Fixed]**

Line 186
* * *
**TC10 4 185 The sentence reads difficult, there seems to be a word missing. Consider to rephrase: "C1 and C2 are the correction factor and offset for scene dependent effects"**

**[Fixed]**

**C1 and C2 are the correction factors for scene-dependent effects and offsets, respectively**

Lines 206-207
* * *
**TC11 4 203 represented ->represent**

**[Fixed]**

Line 220
* * *
**TC12 4 206 1nm-width wave -> 1nm wide wavelength intervals**

**[Fixed]**

Line 223
* * *
**TC13 5 276 indicating a _____ across (word missing).**

**[Fixed]**

**TC14 5 Fig. 6- 10 Figures 6, 8, 9 and 10 should have less white around them and the text should be larger. If you change the aspect ratio and scale to the actual data points, it would be easier to read.**

[Figure]

**The figures have been updated in the revised manuscript. Further checks for the colorblind effects have been applied to all figures.**
* * *
**TC15 5 315 ))_ and (space missing)**

[Figure]
* * *
**TC16 5 Fig 10 Caption: panels g to i are not described**

**[Fixed]**

**The caption is updated by adding details for the missing subplots (g-i)**
* * *
**TC17 6 328 of the TF**

**[Fixed]**
* * *
**TC18 All All Please reconsider when you use the past tense, for example cross calibration is (not was) important.**

[Figure]

**The manuscript will be revised several times to check for further typos and grammatical issues.**
* * *
**TC19 Fig 6/10 When plotting the NIR data vs wavelength, please consider a interrupted scale and focus the plot more on where you do use data**

**[Fixed]**

**We have clarified the way we derive the TFs for Band 4 (NIR). To avoid the misunderstanding, we present the TFs of Band 4 for the whole spectrum range. This is what we have made exactly. Therefore, we have also clarified and differentiated between the derived TFs of Band 2B and 3 vs. Band 4.**

---

## Author Comment (AC2)

Response to comments on egusphere-2025-4942 Submitted 06 Oct 2025

**Anonymous Referee #2**

Referee comment on "**Cross-calibration of GOME and SCIAMACHY Spectrometers Enhanced by Polarization Monitoring Devices Data**" by  Owda, A., Coldewey-Egbers, M., Slijkhuis, S., Lichtenberg, G., and Aberle, B., https://doi.org/10.5194/egusphere-2025-4942, 2025.

**We are grateful to the anonymous referee for taking the time to review our manuscript and provide us with constructive comments. We have carefully considered all comments and suggestions and thoroughly revised the manuscript.**

**In this response letter, we intend to address all comments, concerns, and suggestions point by point. The response will be highlighted in blue and the corresponding modification in the revised manuscript in orange. The original comment from the anonymous referee will be in black.**

**General comments:**

Owda et.al. describe a method to harmonize the reflectances of the spectrometers GOME and SCIAMACHY. Goal is a harmonized time-series of both instruments for reflectances. The authors uses measurements over PICS sites with co-located reflectance measurements from both instruments. The method includes utilizing the spatially higher resolved measuerments of the PMDs of both instrument to ensure homogenous scenes used for the harmonization.

In general, the paper is well written. The ideas behind the method are well described.

Long term data records are an important topic in atmospheric science. Harmonizing the data from different instrument is especially an issue for satellite based measurements. Harmonizing reflectances is a new approach to harmonize the measurements on the spectrometer level.

This paper describes a method to harmonize the reflectances of the satellite instruments GOME and SCIAMACHY. A method to harmonize the irradiance is not covered. This is not

clearly stated in the abstract (according the the abstract, the FDR4ATMOS project aims to provide also irradiance). Please clarify this in the abstract, for example in line 7: This paper presents, for the first time, the cross-calibration methodology for *the reflectance of t*he spectrometers used in the FDR4ATMOS project. You might also consider adding the term *reflectance* to the title of the paper.`

**Response: Thank you very much for the general comments and the positive impression about the paper. We agree that we should highlight clearly the harmonized parameter in the manuscript. We presented only reflectance in the manuscript; therefore, we have added that clearly in the title of the paper and in the abstract as well.** [Fixed]

**See Title "Reflectance-Based Cross-calibration…."**

**See Line 8, "across-calibration methodology for the reflectance of the spectrometers."**

**In the introduction, I miss the motivation for the selected spectral windows. This is driven by the common spectral windows, which contain spectral absorbtions of important trace gases. I suggest adding the relavant trace gases for each band also in Table 2.** [Fixed]

**Response: The motivation for the selection of these ranges of spectral bands is added to the introduction. The relevant trace gases for each band are added in Table 4**

**See Table 4 and**

**Lines 81-83. "The focus of the FDR4ATMOS was on three spectral windows in the ultraviolet, visible, and near-infrared (UV/VIS/NIR), as these regions contain well-characterized absorption features of many key atmospheric trace gases (see Table 4)."**

**In Figure 7, transfer functions (TFs) for the whole band 4 (including O2A) are shown (dashed line). In Figure 10, only the two separate TFs outside the absorbtion are shown. The TF for band 4 needs some clarification:**

**How is the TF for the whole channel build?** [Fixed]

**We agree with your point of view that further clarification is needed.**

We considered only the ratios from the wavelength intervals to the left and to the right of the $O_2$ A-band. The $O_2$ A-band was excluded from the derivation process of TF of Band 4 (NIR). The TF of band 4 is the average of the ratios from the left and right sides of the $O_2$ A-band. Fig 6(g- i) shows the left and right sides of the ratio of the $O_2$ A-band.

Which TF will be used for the harmonization? This is finally stated in the conclusion: `constant value for the whole band`. This needs to be clarified already here. Therefore, the whole channel 4 TFs need to be added to Figure 10, because that is the used one. **[Fixed]**

That is correct; we agree on that. The manuscript is modified as well.

The TF is a constant value that is used to update the whole band (including the O2 A-band). Figure 10 is corrected.

Lines 303-305. For Band 4, the TF was obtained by averaging the ratios on the left and right sides of the $O_2$ A-band, yielding a constant value of 0.93, which represents the TF for Band 4, including the $O_2$ A-band (blue dashed line in Fig. 7c).

Related to this: In Table 1, for the NIR the wavelength intervalls are given as 756–757 & 773–774 nm. These are the intervall used for the TF calculation, but the FDR4ATMOS product will contain the intervall 756-774nm including the O2A Band. Here you need to distinguish between the spectral window in the harmonized product and the windows used in your calculations. You also need to descripe the calculation of the final TF for band 4. **[Fixed]**

We agree. We distinguish between the spectral ranges used to derive the TFs and the harmonized ranges.

See Lines 233-235. Additionally, the note was added to the caption of Table 4

For the derivation of the TF of the NIR spectral window, the $O_2$ A-band is excluded due to its high atmospheric sensitivity and the resulting strong variability in the observed reflectance. Instead, the two wavelength intervals of 1 nm width located immediately to the left and right of the $O_2$ A-band were used to compute the TFs. The ratio measurements obtained in these adjacent intervals were averaged to derive a single wavelength-independent TF, which is then applied to the entire spectral window, including the $O_2$ A-band.

**Specific and technical comments**

**Response: Thank you very much for the specific comments. We believe that your comments were very useful, and fixing these comments helps to improve the readability of the paper.**

**We have dealt with your comments one by one as follows:**

1. **p5, l120:**

    **The SCIAMACHY Level1 product does not contain cloud information. Probably, your cloud information comes from the corresponding Level 2 products. Please clarify.**

    **[Fixed]**

    **Line 124. "For SCIAMACHY, we added cloud information from Level 2 to the standard Level 1c products."**
* * *
2. **p5, 122:**

    **The cluster concept refers to a subdivision of a channel containing a specific wavelength region and detector exposure time, aiming to identify certain important spectral windows in the data.**

    **The goal is not to identify important windows. The goal is to optimize the data rate towards the important spectral windows:**

    **..., aiming to optimize the data rate towards important spectral windows in the data.**

    **We have decided to take this part of the information out of the manuscript. The revised manuscript will not have any description of the cluster concept. The conversion process of Level 1b -> Level 1c was disregarded. The input for the harmonization starts with SCIAMACHY data, which is at Level 1c.**
* * *
3. **p5, l127:**

   with the scial1c tool developed by DLR (the link to the tool is under Section 8).

   I suggest to change this to a normal reference and add the URL to the references, something like: with the scial1c tool developed by DLR (ESA, 2025c).

   **[Fixed]**

   **We have followed your suggestion by putting the links under the references.**
* * *
4. **p7, Fig. 2:**

   Add to the text in the building blocks the variable names used in text/formulas, just as in the first block with ($R_g$) and ($R_s$)

   - SCIA2GOME Pseudo reflectance ->  SCIA2GOME Pseudo reflectance ($R_{SCIA2GOME}$)

   - Ratio of refectance (SCIAMACHY/GOME) ->  Ratio of refectances (Ratio)

   - Transfer functions  ...  -> Transfer functions (TF) ...

   I suggest to add a block for "Outlier removal", that step is missing in the sketch.

   **[Fixed]**

   **Fig.2  has been updated in the revised manuscript based on reviewers' and your feedback.**
* * *
5. **p7, eq(1):**

   Please use a proper multiplication sign in equation (1), this seem to  be a simple dot (.).

   Should be \cdot  in Latex math mode. Same for all further equations.

   **[Fixed]**

   **See equation 1**

6. **p7, Figure 3:**

There is a gap between subsequent SCIAMACHY scanlines. This is not expected; there is (almost) no gap between the scanlines (similar to the two GOME ground pixels in the plot). Please check your figure.

This gap refers to considering only forward scanning in the cross-calibration. The backscan is not considered in the cross-calibration, therefore, there is a gap between the forward scanning of SCIAMACHY.

We have updated the caption with this piece of information.

See caption of Fig 3
* * *
7. **p9, Table2:**

This table is a duplication of Table 1, only provide and refer to Table 1.

[Fixed]

There was a duplicate. Tables 1 and 2 were the same. In the revised manuscript, there are no more duplicated tables.
* * *
8. **p 24, l397:**

the spectral channels with strong absorption features: Only the O2A absorbtion band is excluded, so I suggest to clarify here, that the O2A window is the one excluded.

[Fixed]

O2 A-band was the only window excluded from the derivation of TFs.

See Line 410-411
* * *
9. **p 24, l401:**

*It found that... -> It has been found that...*

[Fixed]

See Line 415

10. 25, Section 8:

Instead of a list of links (with wrong indentations, use the `itemize` environment for lists), make a short text and move the links to the references. Something like:

*SCIAMACHY and GOME Level-1 data are available from ESA (ESA, 2025c; ESA, 2025d),* etc.

[Fixed]

Instead of using links. We moved all links to the reference and cited them as you suggested.

Section 8